# How COVID-19 Pandemic Has Influenced Public Interest in Foods: A Google Trends Analysis of Italian Data

**DOI:** 10.3390/ijerph20031976

**Published:** 2023-01-20

**Authors:** Andrea Maugeri, Martina Barchitta, Vanessa Perticone, Antonella Agodi

**Affiliations:** 1Department of Medical and Surgical Sciences and Advanced Technologies “GF Ingrassia”, University of Catania, 95123 Catania, Italy; 2Department of Economics and Business, University of Catania, 95129 Catania, Italy

**Keywords:** diet, nutrition, dietary risk, COVID-19, SARS-CoV-2, pandemic

## Abstract

Controversy exists about the impact of the COVID-19 pandemic on dietary habits, with studies demonstrating both benefits and drawbacks of this period. We analyzed Google Trends data on specific terms and arguments related to different foods (i.e., fruits, vegetables, legumes, whole grains, nuts and seeds, milk, red meat, processed meat, and sugar-sweetened beverages) in order to evaluate the interest of Italian people before and during the COVID-19 pandemic. Joinpoint regression models were applied to identify the possible time points at which public interest in foods changed (i.e., joinpoints). Interestingly, public interest in specific food categories underwent substantial changes during the period under examination. While some changes did not seem to be related to the COVID-19 pandemic (i.e., legumes and red meat), public interest in fruit, vegetables, milk, and whole grains increased significantly, especially during the first lockdown. It should be noted, however, that the interest in food-related issues returned to prepandemic levels after the first lockdown period. Thus, more efforts and ad hoc designed studies should be encouraged to evaluate the duration and direction of the COVID-19 pandemic’s influence.

## 1. Introduction

In the last few decades, efforts have been made to disentangle the complex relationship between human nutrition and the risk for chronic noncommunicable diseases (NCDs) [1,2,3,4,5,6,7,8,9,10,11,12,13]. In 2017, as a part of the Global Burden of Diseases, Injuries, and Risk Factors Study (GBD), the GBD 2017 diet collaborators demonstrated that dietary risks were responsible for more deaths than any other risk factor, accounting for nearly 11 million deaths and 255 million Disability-Adjusted Life Years (DALYs) globally [14]. The GBD group showed that the main dietary risk factors were diets high in sodium and low in whole grains, fruits and vegetables, nuts and seeds, and omega-3 fatty acids [14]. Overall, these findings highlighted the urgent need to raise public interest in the risks and benefits associated with dietary factors, with the long-term goal of improving human diets across countries. According to Agenda 2030, in fact, nutrition plays a crucial role in achieving the Sustainable Development Goals (SGDs) in general, and specifically in promoting health throughout the life course. There is a specific goal related to nutrition—ending hunger, achieving food security, and promoting sustainable agriculture—but SDGs 4 and 10 also emphasize the importance of promoting healthy nutrition throughout life for all vulnerable groups and groups in need [15].

Throughout the last two years, however, the COVID-19 pandemic has impacted our lives and daily habits, putting away the possibility to achieve the SDSs described earlier. In fact, while our routine has almost returned to normal, some changes have been incorporated. Every sector of our economy has been affected by the pandemic, and food purchases have also been impacted. While restaurant spending dropped in 2020, domestic food consumption increased, especially during the first lockdown [16]. On the other hand, however, fresh food consumption may have declined in favor of highly processed foods due to limited access to daily shopping [17,18]. For these reasons, there is controversy regarding the impact of the COVID-19 pandemic on dietary habits, with studies demonstrating both benefits and drawbacks of this period [17,18,19,20,21,22].

Over the past decade, the internet has become an important resource for health informatics since the increasing availability of online sources provides data for analyzing and predicting human behaviors [23,24,25,26,27,28,29,30,31,32]. Accordingly, two concepts were coined to define the potential use of the internet in public health: the term “Infodemiology”, defined by Eysenbach as “the science of distribution and determinants of information in an electronic medium, specifically the internet, or in a population, with the ultimate aim to inform public health and public policy” and the term “Infoveillance”, which refers to “the longitudinal tracking of infodemiology metrics for surveillance and trend analysis” [33]. In this framework, Google Trends—a free and public online portal of Google Inc. (CA, USA)—allows for the analysis of the Google search activities of people while also considering geospatial and temporal trends for user-specified terms [34]. In light of this, recent studies have proposed Google Trends as a useful tool for analyzing internet search behaviors and, consequently, evaluating public interest in diet-related topics [16,35,36,37].

Therefore, our general hypothesis was that the analysis of Google Trends could help us monitor the public interest in dietary factors and foods. We focused on Italian people because Italy was one of the countries that was most negatively affected by the COVID-19 pandemic [38,39]. As of 2022, Italy was estimated to have 58.9 million residents, distributed throughout 20 regions. According to the last 2021 SDG report “Statistical Information for 2030 Agenda in Italy”, the proportion of households with a fixed/mobile broadband was nearly 78%, while people aged 6 and over who regularly used the internet accounted for approximately 70%. Internet use was more common among men than women, and among individuals with a high educational level [40]. In this scenario, we analyzed Google Trends data on specific terms and arguments related to different foods in order to evaluate the interest of Italian people before and during the COVID-19 pandemic.

## 2. Materials and Methods

### 2.1. Selection of Food-Related Terms

The GBD defined 15 foods/nutrients as risk factors for different disease endpoints, in accordance with the criteria of the importance of the risk factor, availability of sufficient data, and strength of epidemiological evidence supporting the health-related effects [14]. Among them, we selected the following Google Trends topics that reflected internet search activities on nine foods: fruits; vegetables, legumes, whole grains, nuts and seeds, milk, red meat, processed meat, and sugar-sweetened beverages. In our analysis, we did not consider the dietary risk factors related to nutrients (i.e., fiber, calcium, omega-3 fatty acids, polyunsaturated fatty acids, trans fatty acids, and sodium) because they showed low levels of public interest.

### 2.2. Google Trends Data

Google Trends data were obtained from the website https://trends.google.com/trends on 15 July 2022. Data were referred to Google searches carried out in Italy for a 234-week period (i.e., from 1 January 2018 to 30 June 2022). For each term, data were provided as relative search volume (RSV), which was obtained by dividing the number of searches for a given term by the total number of searches. This normalization process resulted in data ranging from 0 (i.e., the lowest RSV) to 100 (i.e., the highest RSV). The adjustment process also excluded queries that were made over a short time frame from the same internet protocol address and queries that contained special characters.

### 2.3. Statistical Analysis

After the initial dataset exploration, univariate analyses were conducted using RStudio. Google Trends data were summarized using the median and interquartile range (IQR) due to their skewed distributions. Next, joinpoint regression analyses were carried out using the Joinpoint Regression Program (version 4.3.1.0; Statistical Research and Applications Branch, National Cancer Institute, Bethesda, MD, USA) provided by the Surveillance, Epidemiology, and End Results Program (National Cancer Institute) on the website http://surveillance.cancer.gov/joinpoint (accessed on 1 August 2022). Joinpoint regression is suitable for analyzing the time series of rates, proportions, or any other measure to identify the possible time points at which a trend changes (i.e., joinpoints). Joinpoint regression differs from other similar models (e.g., piecewise or segmented regressions) because it has the constraint of continuity at the change points. Moreover, the choice of the number of joinpoints and their locations are estimated within the model. In the current study, the joinpoint regression model was applied on the logarithmic transformation of the outcome variable, with 5000 permutations and assuming uncorrelated errors. As suggested by Lerman, the grid search method was used to determine where to locate the joinpoints on the timescale [41]. The results are expressed as the Weekly Percent Change (WPC), which reflects the average weekly percentage change per week between different joinpoints. All statistical analyses were performed with a significance level of 0.05.

## 3. Results

Figure 1 shows the public interest of Italian people in the nine food categories under investigation from 1 January 2018 to 30 June 2022. Overall, the RSV was the highest for processed meat (median = 56%; IQR = 50–67%) followed by vegetables (median = 52%; IQR = 48–56%) and whole grains (median = 50%; IQR = 43–56%). The RSV was the lowest for legumes (median = 13%; IQR = 7–20%) followed by milk (median = 31%; IQR = 23–43%) and nuts and seeds (median = 34%; IQR = 30–38%).

Public interest in fruit, vegetables, milk, and whole grains exhibited a similar trend, with the highest peak observed during the first COVID-19 lockdown (Figure 2 and Figure 3). The first lockdown in Italy, in fact, began on 21 February 2020, covering ten municipalities in the regions of Lombardy and Veneto. The quarantine zone was further expanded to cover much of Northern Italy on 8 March 2020 and the entire country on 9 March 2020.

Except for slight seasonal variations, public interest in fruit was stable from 1 January 2018 to 1 March 2020, which corresponded to the first joinpoint of the time series (113th week; 95% CI = 110–114). Public interest in fruit then increased (WPC = +37.13%) up to the highest peak reached during the week from 15 to 22 March 2020 (i.e., the second joinpoint detected by the analysis on the 116th week; 95% CI = 115–119). Finally, public interest decreased (WPC = −3.39%) until 30 August 2020 (i.e., the third joinpoint on the 139th week; 95% CI = 130–149) and then stabilized (Figure 2A).

Even the interest in vegetables was stable until 1 March 2020 (i.e., the first joinpoint of the time series on the 113th week; 95% CI = 110–115). Next, it increased (WPC = +20.21%) up to the highest peak reached during the week from 22 to 29 March 2020 (i.e., the second joinpoint on the 117th week; 95% CI = 115–120). Finally, public interest decreased (WPC = −7.07%) until 24 May 2020 (i.e., the third joinpoint on the 125th week; 95% CI = 122–131) and then stabilized (Figure 2B).

As with fruit and vegetables, even the public interest in milk remained stable until the first national lockdown; the first joinpoint of the time series, in fact, was detected on the 112th week (95% CI = 108–114th). Next, the interest increased (WPC = +36.38%) from 23 February to 22 March 2020 (i.e., the second joinpoint detected by the analysis on the 116th week; 95% CI = 114–119th). Finally, it decreased (WPC = −8.12%) until 21 June 2020 (i.e., the third joinpoint on the 129th week; 95% CI = 125–132nd), and then stabilized (Figure 3A).

The trend was similar for whole grains, even if the public interest reached its highest peak slightly after. In fact, the interest was stable until 29 December 2019 (i.e., the first joinpoint of the time series on the 104th week; 95% CI = 100–114th) and then increased (WPC = +4.99%), reaching the peak during the week from 12 to 19 April 2020 (i.e., the second joinpoint on the 120th week; 95% CI = 114–124th). Finally, public interest decreased (WPC = −9.30%) until 31 May 2020 (i.e., the third joinpoint on the 126th week; 95% CI = 122–134th) and then stabilized (Figure 3B).

Public interest in legumes exhibited a peak that, however, was not associated with the COVID-19 pandemic. The interest, in fact, was stable from 1 January 2018 to 16 December 2018, which corresponded to the first joinpoint of the time series (50th week; 95% CI = 43–58th). It then increased (WPC = +24.28%) up to the highest peak reached during the week from 10 to 17 March 2019 (i.e., the second joinpoint detected by the analysis on the 63rd week; 95% CI = 54–65th). Finally, public interest rapidly decreased (WPC = −55.81%) until 7 April 2019 (i.e., the third joinpoint on the 66th week; 95% CI = 63–81st) and then stabilized (Figure 4A).

The analysis of public interest in red meat also detected three joinpoints that, however, were differentially located in the time series. The interest in red meat, in fact, exhibited a slight increase (WPC = 0.16%) from 1 January 2018 to 27 December 2020 (i.e., the first joinpoint detected by the analysis on the 156th week; 95% CI = 132–179), which became more marked (WPC = 14.97%) up to the highest peak reached during the week from 10 to 17 January 2021 (i.e., the second joinpoint on the 159th week; 95% CI = 156–188th). Next, public interest decreased (WPC = −2.45%) until 27 June 2021 (i.e., the third joinpoint on the 182nd week; 95% CI = 173–230th) and then increased again (WPC = 0.55%; Figure 4B).

The analyses of public interest in nuts and seeds, processed meat, and sugar-sweetened beverages showed seasonal variations, without significant joinpoints detected from 1 January 2018 to 30 June 2022 (Appendix A).

## 4. Discussion

In the present study, we demonstrated that public interest in specific food categories underwent substantial changes during the period under examination. While some changes did not seem to be related to the COVID-19 pandemic (i.e., legumes and red meat), public interest in fruit, vegetables, milk, and whole grains changed significantly, especially during the first lockdown. In fact, the COVID-19 pandemic has not only impacted healthcare systems, but also global economies, world trade, tourism, and social restrictions. As a result of these restrictions, people have changed their habits and moods, their purchasing behaviors, and their interest in issues related to food security and food waste [16,17,18,19,20,42]. During the first lockdown, people faced a completely new situation and began to become interested in which elements might be healthier in fighting the virus. A similar Google Trends analysis by Mayasari and colleagues demonstrated that during the COVID-19 lockdown, there was an increased interest in keywords related to food security, especially in the USA, Canada, the UK, Germany, Ireland, Australia, New Zeeland, and Singapore. In particular, public interest in eating away from home decreased, while food delivery increased. This was also accompanied by an increased interest in indoor behavior-related terms (e.g., playing videogames, watching TV, recipe, and cake) at the expense of outdoor-related terms (e.g., hotel, cinema, park, and resort). Interestingly, the authors also found positive correlations between daily confirmed COVID-19 cases and search volumes of terms related to food security, dietary behaviors, immune-related nutrients/herbs/food, and negative correlations with terms related to outdoor activities [16]. An important contribution was also made by Hamulka and colleagues about the interest in dietary supplements during the first two waves of the COVID-19 pandemic. In particular, the authors reported a great interest in immune-related compounds and foods, such as vitamins C and D, zinc, omega-3, garlic, ginger, and turmeric [36]. Nucci and colleagues instead focused on the most searched terms related to diet and dietary patterns. In general, the Mediterranean diet was the most frequently searched diet, followed by the Pescetarian, Macrobiotic, and Ketogenic diets. Low interest was instead evident for the Atkins, Intermittent-fasting, Scarsdale, Vegan/Vegetarian, and acid–base diets. However, the authors reported an increased interest in the Ketogenic, FODMAP, and Intermittent-fasting diets after the beginning of the COVID-19 pandemic, while interest in the Vegan/Vegetarian, Atkins, Kousmine, and Fruitarian diets decreased [35].

The potential impact of the COVID-19 pandemic on dietary choices and food-related issues has also been confirmed by observational studies and surveys, even if the direction of this effect remains controversial. A previous study reported how changes in dietary habits occurred in some European countries during the pandemic [43]. Further research has shown that the pandemic also affected food purchasing, preparation, and food waste [44,45,46]. The scoping review by Bennet and colleagues summarized both the negative and positive effects of lockdown procedures on dietary habits in Europe and globally [17]. While some studies found an increase in the consumption of so-called comfort foods (e.g., sweets, snacks, fried and processed foods), others reported favorable changes, such as an increase in the consumption of fresh foods [17]. In particular, results from Italian studies suggested an increase in home cooking [18,47] but also a higher consumption of comfort foods [48,49,50]. These results, along with those produced through the analysis of Google Trends, suggested that the COVID-19 pandemic certainly influenced the interest in food-related issues and consumers’ behaviors. The high public interest in fruits, vegetables, milk, and whole grain during the lockdown may be attributed to the increased home cooking and consumption of fresh foods. However, it remains unclear whether long-term effects of the pandemic will be evident. The monitoring of dietary habits might be too time consuming and expensive for epidemiological studies, whereas Google Trends could provide evidence rapidly and at a low cost. In fact, our study was useful to demonstrate that the interest in food-related issues returned to prepandemic levels after the first lockdown period. Further studies—even those combining the methodology that we used with ad-hoc surveys—are necessary to solve controversies about the benefits and drawbacks of the COVID-19 pandemic on dietary choices.

Some limitations should be considered when interpreting our results. First, Google Trends data are a proxy measure of the public interest in specific themes, since they reflect the interest of those who have access to internet. Second, Google Trends data indicate only the search volume about specific terms, and not people’s feelings and opinions. This resulted in a mere quantitative analysis that did not provide information on the direction in which the pandemic affected public interest in food categories. Third, our analysis did not take into account the sociodemographic characteristics of internet users; therefore, variability among different demographic factors could not be determined. Moreover, it would be interesting to investigate whether differences exist between Italian regions. It was not possible, however, to obtain data on public interest over time at the regional level or to analyze internet users based on their geographical location. Finally, additional food-related terms could be investigated, such as those related to prevention, diseases, risks, and food safety. Although we focused on foods that constituted well-established dietary risks, findings might vary according to the different terms used. An innovative approach to integrate this information and hence make it possible to better evaluate the impact of the COVID-19 pandemic is represent by the use of mobile apps for the collection of dietary data in real time and in the real world [21,51,52].

Our work also has several strengths. First, it represented a comprehensive analysis of public interest in foods, considering nine of the fifteen factors that were defined as the main dietary risk factors by the GBD [14]. Second, the analysis covered a long period of time spanning from 2018 to 2022 in order to monitor public interest before and during the COVID-19 pandemic. Third, we applied a joinpoint regression analysis that allowed us to analyze time-series data and identify the possible time points at which the trends changed without a priori assumptions.

## 5. Conclusions

In conclusion, our Google Trends analysis of Italian data showed that public interest in specific food-related terms increased as a consequence of the COVID-19 pandemic, and especially during the first national lockdown. However, more efforts and ad hoc designed studies should be encouraged to evaluate the duration and direction of these effects.

## Figures and Tables

**Figure 1 ijerph-20-01976-f001:**
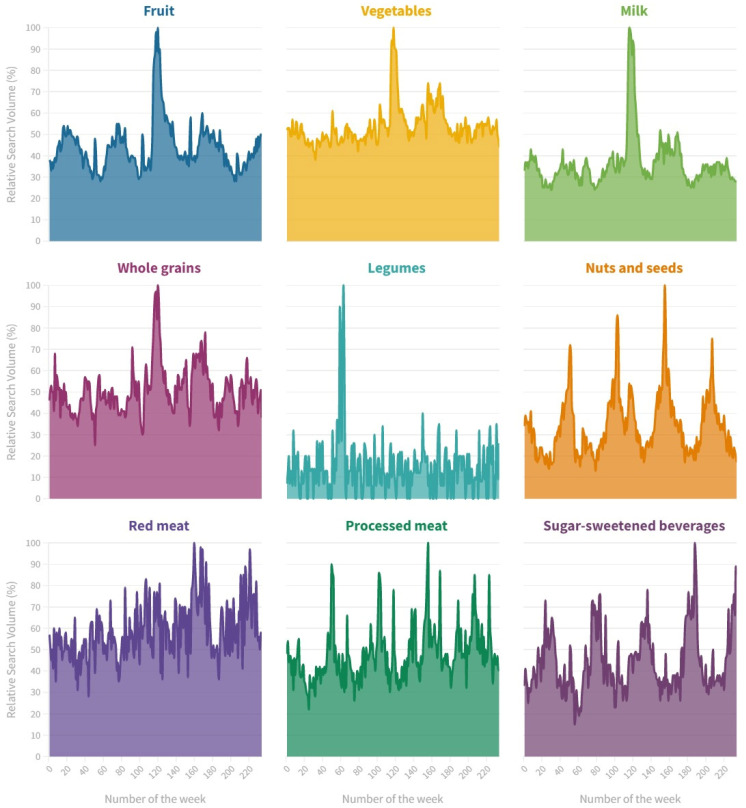
Relative search volumes of food-related terms from 1 January 2018 to 30 June 2022.

**Figure 2 ijerph-20-01976-f002:**
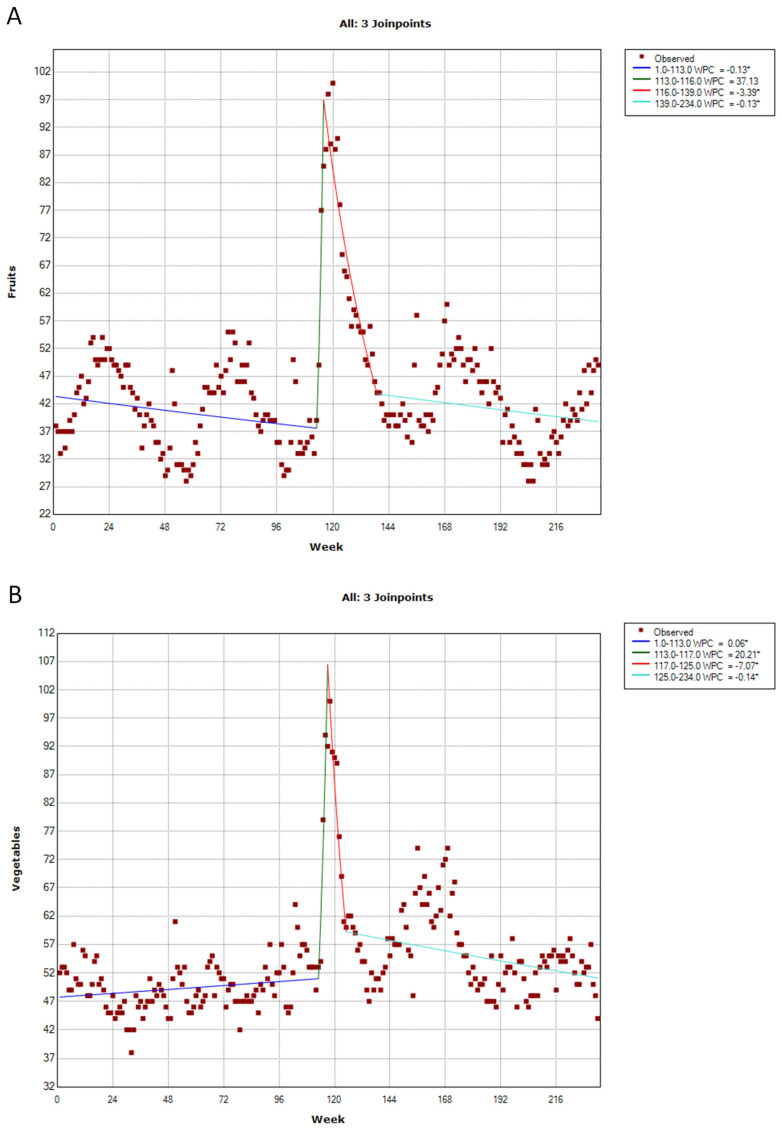
Joinpoint analysis of public interest in fruit (**A**) and vegetables (**B**). * *p*-value < 0.05.

**Figure 3 ijerph-20-01976-f003:**
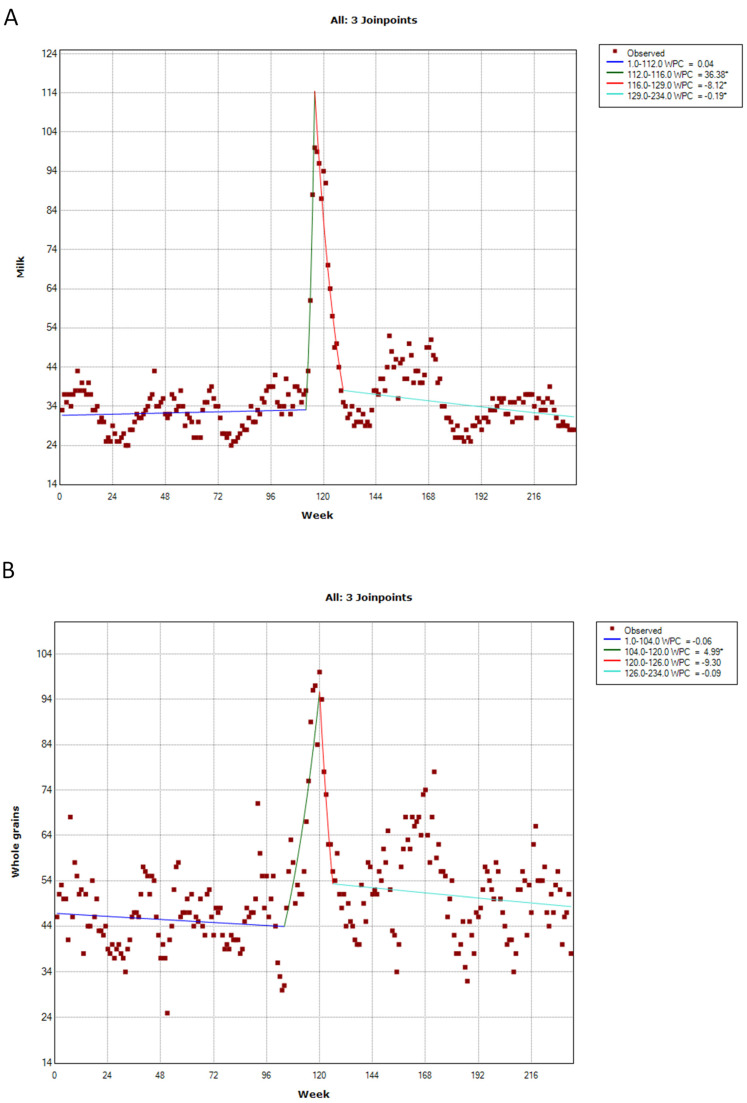
Joinpoint analysis of public interest in milk (**A**) and whole grains (**B**). * *p*-value < 0.05.

**Figure 4 ijerph-20-01976-f004:**
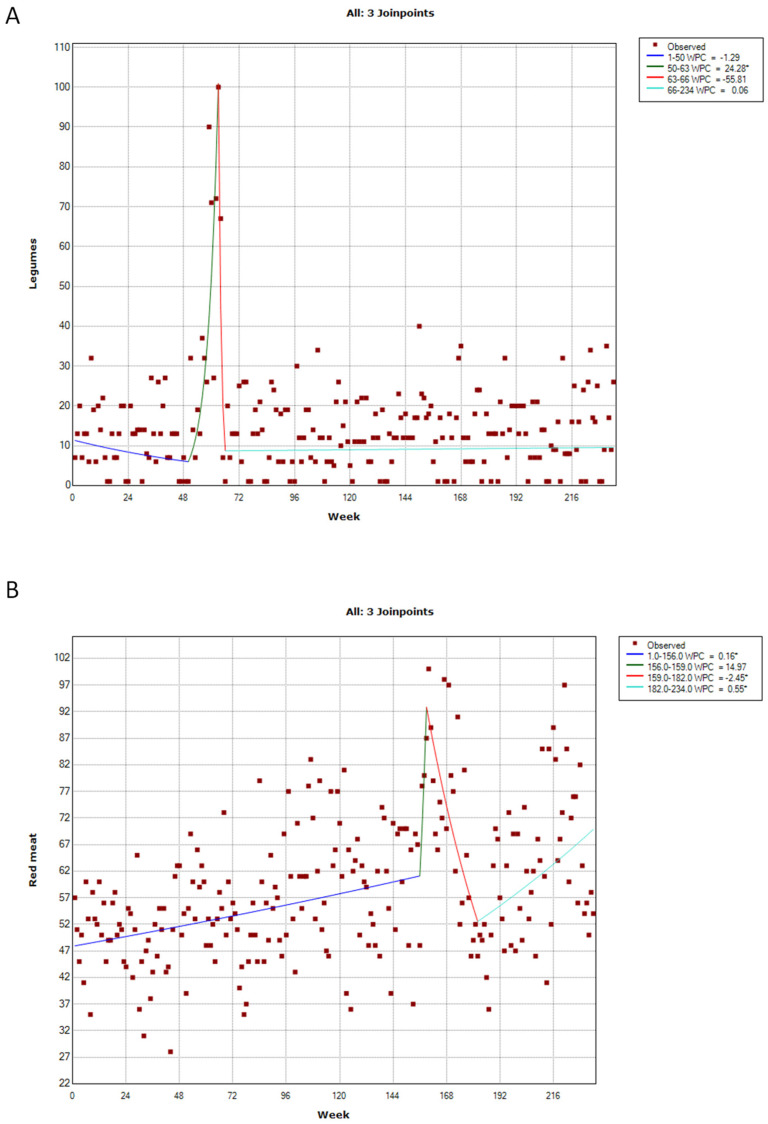
Joinpoint analysis of public interest in legumes (**A**) and red meat (**B**). * *p*-value < 0.05.

## Data Availability

The data that support the findings of this study are available from the corresponding author upon reasonable request.

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
