# Peer review of "How COVID-19 Pandemic Has Influenced Public Interest in Foods: A Google Trends Analysis of Italian Data"

_ijerph, 2023, doi:10.3390/ijerph20031976_

Round 1
Reviewer 1 Report
The manuscript is well-written, however I have a few concerns.
Scientific papers using g t take into account data from across the continent or even the world. Therefore, they provide valuable data. In case the Authors narrow the analysis to only one country, it is necessary to link the data obtained with the surveys that focus on the frequency of consumption and consumers' purchase preferences. It would also be worthwhile to analyze the data by region of the country.
I also ask the Authors to explain in more detail what their research provides to science and public health care that is new.
Author Response
Dear Editor,
Please consider the revised version of the manuscript entitled “How COVID-19 pandemic has influenced public interest in foods: a Google Trends analysis of Italian data” in which we have considered all comments and suggestions from reviewers. This letter is intended for the convenience of the editor and reviewers and contains the list of the requested changes. The following list of changes and answers to comments of Reviewers addresses all revisions made in the manuscript (in red font).
Reviewer 1
R: The manuscript is well-written, however I have a few concerns.
Answer: We would like to take this opportunity to thank the Reviewer for his/her comments and suggestions which helped us in improving our manuscript.
R: Scientific papers using g t take into account data from across the continent or even the world. Therefore, they provide valuable data. In case the Authors narrow the analysis to only one country, it is necessary to link the data obtained with the surveys that focus on the frequency of consumption and consumers' purchase preferences. It would also be worthwhile to analyze the data by region of the country.
A: As suggested, we included in the discussion section a paragraph describing the impact of COVID-19 pandemic on dietary habits and consumers’ behaviours. We also recognize that it would be interesting to analyze Google Trends data at the regional level. However, as now described in the limitation section, it was not possible to obtain data on public interest over time at the regional level.
R: I also ask the Authors to explain in more detail what their research provides to science and public health care that is new.
A: As part of the discussion section, we have provided a commentary on the results of our study within the context of previous research, as well as the importance in the current pandemic scenario.
Reviewer 2 Report
The work titled “How COVID-19 pandemic has influenced public interest in foods: a Google Trends analysis of Italian data” gives some interesting results on the effect COVID-19 has brought regarding public interest in foods. The study will be useful to the readers, especially as COVID-19 remain with us.
I have some comments as follows that should be taken into consideration.
(1) I think this study should extensively compare findings from the methodology used and those involving questionnaire surveys. By doing so, you will find explanations for some trends in public interest regarding foods. For instance, authors reported a significant increase in public interest in fruit, vegetables, milk, and whole grains, especially during the first lockdown. Scientific reasons for this can be found in many other related studies in Italy or elsewhere. Authors should consider the following studies and search for more to improve the current discussion.
https://doi.org/10.3390/su13073702
https://doi.org/10.18461/ijfsd.v13i1.A1
https://doi.org/10.1016/j.appet.2021.105110
https://doi.org/10.3390/su13063381
(2) Other limitations of the study to be included include the lack of data regarding the demographics of the expressing their interest. Therefore, variability among different demographic factors cannot be determined.
Author Response
Dear Editor,
Please consider the revised version of the manuscript entitled “How COVID-19 pandemic has influenced public interest in foods: a Google Trends analysis of Italian data” in which we have considered all comments and suggestions from reviewers. This letter is intended for the convenience of the editor and reviewers and contains the list of the requested changes. The following list of changes and answers to comments of Reviewers addresses all revisions made in the manuscript (in red font).
Reviewer 2
R: The work titled “How COVID-19 pandemic has influenced public interest in foods: a Google Trends analysis of Italian data” gives some interesting results on the effect COVID-19 has brought regarding public interest in foods. The study will be useful to the readers, especially as COVID-19 remain with us.
Answer: We would like to take this opportunity to thank the Reviewer for his/her comments and suggestions which helped us in improving our manuscript.
R: I have some comments as follows that should be taken into consideration. I think this study should extensively compare findings from the methodology used and those involving questionnaire surveys. By doing so, you will find explanations for some trends in public interest regarding foods. For instance, authors reported a significant increase in public interest in fruit, vegetables, milk, and whole grains, especially during the first lockdown. Scientific reasons for this can be found in many other related studies in Italy or elsewhere. Authors should consider the following studies and search for more to improve the current discussion.
https://doi.org/10.3390/su13073702
https://doi.org/10.18461/ijfsd.v13i1.A1
https://doi.org/10.1016/j.appet.2021.105110
https://doi.org/10.3390/su13063381
A: As suggested, we included in the discussion section a paragraph related to epidemiological studies and surveys on the impact of COVID-19 pandemic on dietary choices. As described, however, findings are still controversial since some studies reported benefits from the pandemic while others demonstrated a negative effect. We also commented previous findings that – at least partially – supported the high public interest observed for fruits, vegetables, milk, and whole grain. In this section, we have also included the suggested references.
R: Other limitations of the study to be included include the lack of data regarding the demographics of the expressing their interest. Therefore, variability among different demographic factors cannot be determined.
A: As suggested, we have included this point as a limitation of our study.
Reviewer 3 Report
The paper does not emphasize the strength, but only the limitations of research. It is necessary to highlight the strenght and show in the discussion that they go beyond the limitation of the study.
The discussion should highlight the potential positive impact that interest rates can have on certain food groups. Did it have an impact on changing eating habits?
A major limitation of the study is that it did not take into account the sociodemographic characteristics of the respondents. Although it is not possible to obtain specific data given the methodology of data collection, it would be good to include the characteristics of the respondents in relation to the area of residence (region). In any case, it is necessary to include some other characteristics of the population in the analysis.
Author Response
Dear Editor,
Please consider the revised version of the manuscript entitled “How COVID-19 pandemic has influenced public interest in foods: a Google Trends analysis of Italian data” in which we have considered all comments and suggestions from reviewers. This letter is intended for the convenience of the editor and reviewers and contains the list of the requested changes. The following list of changes and answers to comments of Reviewers addresses all revisions made in the manuscript (in red font).
Reviewer 3
R: The paper does not emphasize the strength, but only the limitations of research. It is necessary to highlight the strength and show in the discussion that they go beyond the limitation of the study.
Answer: We would like to take this opportunity to thank the Reviewer for his/her comments and suggestions which helped us in improving our manuscript. As suggested, we have added a paragraph on the main strengths of our work.
R: The discussion should highlight the potential positive impact that interest rates can have on certain food groups. Did it have an impact on changing eating habits?
A: As suggested, in the discussion section, we have commented on the potential relationship between public interest in foods and consumers’ behaviours.
R: A major limitation of the study is that it did not take into account the sociodemographic characteristics of the respondents. Although it is not possible to obtain specific data given the methodology of data collection, it would be good to include the characteristics of the respondents in relation to the area of residence (region). In any case, it is necessary to include some other characteristics of the population in the analysis.
A: As suggested, we have included this point as a limitation of our study. Moreover, we have provided a description of the Italian population and its internet activity in the introduction section.
Round 2
Reviewer 1 Report
Dear Authors,
Thank you for considering my comments and putting efforts to improve the manuscript.
Reviewer 2 Report
Based on the revision being made. The manuscript is now acceptable.